Genome-wide identification and analysis of cystatin family genes in Sorghum (Sorghum bicolor (L.) Moench)

Li Jie 1
Liu Xinhao 2
Wang Qingmei 2
Sun Junyan 1
He Dexian hedexian@126.com 3
1 College of Agronomy, Xinyang Agriculture and Forestry University , Xinyang , Henan Province , China
2 Central Laboratory, Xinyang Agriculture and Forestry University , Xinyang , Henan Province , China
3 Collaborative Innovation Center of Henan Grain Crops/National Key Laboratory of Wheat and Maize Crop Science, College of Agronomy, Henan Agricultural University , Zhengzhou , China
Lazo Gerard
Electronic publication date: 2021 Jan 21
Publication date: 2021
Volume: 9
Electronic Location ID: e10617
Received 2019 Dec 30; Accepted 2020 Nov 30
Copyright: ©2021 Li et al.
Copyright year: 2021
Copyright holder: Li et al.
License: This is an open access article distributed under the terms of the Creative Commons Attribution License, which permits unrestricted use, distribution, reproduction and adaptation in any medium and for any purpose provided that it is properly attributed. For attribution, the original author(s), title, publication source (PeerJ) and either DOI or URL of the article must be cited.
License URL: https://creativecommons.org/licenses/by/4.0/

Keywords: Sorghum, Cystatin genes, Expression profiles, Biotic stress, Abiotic stress

Funding: Youth Foundation of Xinyang Agriculture and Forestry University 201701006 Key Scientific Research Projects of Colleges and Universities in Henan Province 20B210019 Key Projects in the National Science and Technology Pillar Program during the Twelfth Five-year-Plan-Period 2013BAD07B National Key Research and Development Program “Science and Technology Innovation of High Grain Production Efficiency” of China 2018YFD0300701 This work was supported by the Youth Foundation of Xinyang Agriculture and Forestry University (201701006), the Key Scientific Research Projects of Colleges and Universities in Henan Province (20B210019), the Key Projects in the National Science and Technology Pillar Program during the Twelfth Five-year-Plan-Period (2013BAD07B), and the National Key Research and Development Program “Science and Technology Innovation of High Grain Production Efficiency” of China (2018YFD0300701). The funders had no role in study design, data collection and analysis, decision to publish, or preparation of the manuscript.

==============================
To set a systematic study of the Sorghum cystatins (SbCys) gene family, a genome-wide analysis of the SbCys family genes was performed by bioinformatics-based methods. In total, 18 SbCys genes were identified in Sorghum, which were distributed unevenly on chromosomes, and two genes were involved in a tandem duplication event. All SbCys genes had similar exon/intron structure and motifs, indicating their high evolutionary conservation. Transcriptome analysis showed that 16 SbCys genes were expressed in different tissues, and most genes displayed higher expression levels in reproductive tissues than in vegetative tissues, indicating that the SbCys genes participated in the regulation of seed formation. Furthermore, the expression profiles of the SbCys genes revealed that seven cystatin family genes were induced during Bipolaris sorghicola infection and only two genes were responsive to aphid infestation. In addition, quantitative real-time polymerase chain reaction (qRT-PCR) confirmed that 17 SbCys genes were induced by one or two abiotic stresses (dehydration, salt, and ABA stresses). The interaction network indicated that SbCys proteins were associated with several biological processes, including seed development and stress responses. Notably, the expression of SbCys4 was up-regulated under biotic and abiotic stresses, suggesting its potential roles in mediating the responses of Sorghum to adverse environmental impact. Our results provide new insights into the structural and functional characteristics of the SbCys gene family, which lay the foundation for better understanding the roles and regulatory mechanism of Sorghum cystatins in seed development and responses to different stress conditions.

Introduction

Cystatins are competitive and reversible inhibitors of cystein proteases from families C1A and C13, which have been identified in many plant species (Martinez & Diaz, 2008; Zhao et al., 2014). Cystatins are categorized into three groups, including stefins without disulfide bonds (Group I), cystatins with four conserved Cys residues forming two disulfide bonds (Group II), and kininogens with repeated, stefin-like domains (Group III) (Meriem et al., 2010). Cystatins are widely distributed in animal and plant systems (Tremblay, Goulet & Michaud, 2019). Based on their primary sequence homology, cystatins contain three signature motifs including a QxVxG reactive site, a tryptophan residue (W) located downstream of the reactive site, and one or two glycine (G) residues in the flexible N terminus of the protein. These three motifs are important for the cystatin inhibitory mechanism (Martinez et al., 2009). In addition, a consensus sequence ([LVI]-[AGT]-[RKE]-[FY]-[AS]-[VI]-x-[EDQV]-[HYFQ]-N) in cystatins is conformed to a predicted secondary α-helix structure (Meriem et al., 2010). Most plant cystatins are small proteins with a molecular mass in the 12- to 16-kD range (Meriem et al., 2010). Some plant cystatins contain a C-terminal extension that raises their molecular weights up to 23 kDa. The longer C-terminal extensions are thought to be involved in the inhibition of cysteine protease activities in the peptidase C13 family (Martinez et al., 2007; Martinez & Diaz, 2008).

The principal functions of plant cystatins are related to the regulation of endogenous cystein proteases during plant growth and development, senescence, and programmed cell death (Belenghi et al., 2010; Díazmendoza et al., 2014; Zhao et al., 2014). Additionally, plant cystatins have been used as effective molecules against different pests and pathogens (Martinez et al., 2016). For example, several publications reported the inhibition of recombinant cystatins on the growth of some pests and fungi (Lima, Dos Reis & De Souza, 2015; Tremblay, Goulet & Michaud, 2019). Tomato plants over-expressing the wheat cystatin TaMDC1 displayed a broad stress resistance to bacterial pathogen, and the defense responses were mediated by methyl jasmonate and salicylic acid (Christova et al., 2018). The inhibition of amaranth cystatin on the digestive insect cysteine endopeptidases was observed by Valdes-Rodriguez et al. (2015). Plant cystatins are also involved in the responses to abiotic stresses, such as over-expression of MpCYS4 in apple delayed natural and stress-induced leaf senescence (Tan et al., 2017). Song et al. (2017) found that the expression of AtCYS5 was induced by heat stress (HS) and exogenous ABA treatment in germinating seed, furthermore, over expression of AtCYS5 enhanced HS tolerance in transgenic Arabidopsis.

To date, cystatin family genes had been well described in several plant species such as Arabidopsis, rice, soybean, wheat, Populus trichocarpa, and Brachypodium distachyon (Martinez & Diaz, 2008; Wang et al., 2015; Yuan et al., 2016; Dutt et al. 2012; Subburaj et al., 2017). However, a genome-wide study of cystatin family genes in Sorghum has not yet been performed. Sorghum is the world’s fifth biggest crop (after rice, wheat, maize, and barley), belonging to a C4 grass that grows in arid and semi-arid regions (Taylor et al., 2010). Its drought tolerance is a consequence of morphological and anatomical characteristics (i.e., thick leaf wax, deep root system) and physiological responses (i.e., stay-green, osmotic adjustment). Hence, Sorghum is an excellent model plant for the study of plant response to drought stress (Sunita et al., 2011). Recently, the completion of the whole genome assembly of Sorghum (Sorghum bicolor L. Moench) makes it possible to identify and analyze cystatin family genes in Sorghum (Paterson et al., 2009). In this study, we aimed to perform a genome-wide identification of SbCys family genes in Sorghum and analyze their phylogeny, conserved motifs, structure, cis-elements, and expression profile in different tissues. We also explored the expression patterns of SbCys genes in response to biotic and abiotic stresses. The results may lay a foundation for further functional analyses of cystatin genes.

Materials and Methods

Identification of SbCys family members in Sorghum genome

The identification of SbCys candidates was conducted according to the methods of Lozano et al. (2015) with some modification. The cystatin sequences of Arabidopsis, rice (Oryza sativa), and barely (Hordeum vulgare) were downloaded from TAIR (http://www.Arabidopsis.org), the Rice Genome Annotation Project (http://rice.plantbiology.msu.edu/index.shtml), and Ensembl database (http://plants.ensembl.org/Hordeum_vulgare/Info/Index), respectively. The whole-genome sequence of Sorghum was downloaded from Ensembl database (http://plants.ensembl.org/Sorghum_bicolor/Info/Index). Then predicted proteins from the Sorghum genome were scanned using HMMER v3 (http://hmmer.org/) using the Hidden Markov Model (HMM) profile of cystatin (PF00031) from the Pfam protein family database (http://pfam.xfam.org/) (Finn, Clements & Eddy, 2011). From the proteins obtained using the raw cystatin HMM, a high-quality protein set with a cut-off e-value <1 × 10−10 was aligned and used to construct a Sorghum specific cystatin HMM using hmmbuild from the HMMER v3 suite. Then all proteins with e-value < 0.01 were selected by the new Sorghum specific HMM. Cystatin sequences were further filtered based on the closest homolog from Arabidopsis, Oryza sativa, and Hordeum vulgare using ClustalW and the UNIREF100 sequence database. Proteins that have no typical domain (Aspartic acid proteinase inhibitor) and reactive site motif (QxVxG) were removed from posterior analysis.

Sequence alignment, structure analysis, and phylogenetic tree construction

The Multiple Expectation for Motif Elicitation (MEME) program was used to identify conserved motifs shared among SbCys proteins. The parameters of MEME were as follows: maximum number of motifs, 10; optimum width, between 6 and 50; and number of repetitions, any.

The three-dimensional structures of Sorghum cystatins were modelled by the automated SWISS-MODEL program (http://swissmodel.expasy.org/interactive). The known crystal structure of rice oryzacystatin I (OC-I) and SiCYS (Hu et al., 2015; Yuan et al., 2016) were used to construct the homology-based models. Structure analysis was conducted by the RasMol 2.7 program.

A phylogenetic tree was constructed using MEGA X with the maximum likelihood method according to the Whelan and Goldman + freq. Model. Bootstrap analysis was performed by 1000 replicates with the p-distance model. The phylogenetic tree was visualized and optimized in Figtree (http://tree.bio.ed.ac.uk/software/figtree/).

Transcript structures, chromosomal location and gene duplication

The genomic structure of each SbCys gene was derived from the alignment of their coding sequence to their corresponding genome full-length sequence. The diagrams of these SbCys genes were drawn by the Gene Structure Display Server (GSDS, http://gsds.gao-lab.org/) (Hu et al., 2014). The chromosomal locations of SbCys genes were retrieved from the Sorghum_bicolor_NCBIv3 map. The genes were plotted on chromosomes using the Map Gene2chromosome (MG2C, version 2.0) tool (http://mg2c.iask.in/). Gene duplication events of SbCys family genes were investigated according to the following two criteria: (1) the alignment covered >75% of the longer gene, (2) the aligned region had an identity >75%, (3) located in less than 100 kb single region or separated by less than five genes. For microsynteny analysis, the CDS sequence of every cystatin from Arabidopsis, barley, rice, and Sorghum was used as the query to search against all other cystatins using NCBI_blast software with e-value ≤ 1e−10. The Circos software was used to display the results of collinear gene pairs (Krzywinski et al., 2009).

Calculation of Ka and Ks

To assess the degree of natural selection on SbCys genes, the rate ratio of Ka (nonsynonymous substitution rate) to Ks (synonymous substitution rate) was calculated using KaKs Calculator 2.0 (Zhang et al., 2006). The Ka/Ks ratio >1, <1, or = 1 indicates positive, negative, or neutral evolution, respectively (Yadav et al., 2015).

Promoter analysis of SbCys genes

To investigate the cis-regulatory elements in a promoter region, the upstream sequences (1.5 kb) of the start codon in each SbCys gene were scanned in the PlantCARE database (http://bioinformatics.psb.ugent.be/webtools/plantcare/html/) and New PLACE (https://www.dna.affrc.go.jp/PLACE/?action=newplace).

Analysis of interaction networks of the SbCys proteins

The functional interacting network models of SbCys proteins were integrated using the web STRING program (http://string-db.org/) based on an Arabidopsis association model; the confidence parameters were set at a 0.40 threshold, the number of interactors was set to five interactors. Arabidopsis AtCys proteins were mapped to Sorghum SbCys proteins based on their homologous relationship. The interaction network of SbCys proteins was drawn by Cytoscape_v3.6.0.

Expression analysis of SbCys genes under biotic stresses

The RNA-Seq data used for investigating the expression patterns of SbCys genes in various tissues were downloaded from the NCBI SRA (Sequence Read Archive) database (ERP024508) (Wang et al., 2018). Root, shoot, and seedling were collected at 14 days after germination. Embryo, endosperm, and pericarp were collected at 20 days after pollination. Pollen samples were collected at booting stage. Inflorescences were collected according to the sizes: 1–5 mm, 5–10 mm, and 1–2 cm. Three biological replicates were performed for each plant tissue. RNA was sequenced using the Illumina HiSeq 2500 system to generate 250 bp pair-end reads.

RNA-seq data of biotic stresses were obtained from two experiments. The first experiment measured the transcriptome response of a resistant Sorghum (Sorghum bicolor L. Moench) infected with Bipolaris sorghicola (Yazawa et al., 2013). RNA samples were collected at 0, 12, and 24 h post-inoculation with one biological replicate. RNA-seq was run using Illumina technology to give 100-base-pair single-end reads on a HiSeq2000 system. The second study measured changes in the transcriptome of Sorghum leaves infested by sugarcane aphid (Tetreault et al., 2019). The RNA-seq data were downloaded from the NCBI SRA database. In this study, two treatments (infested and control) were arranged and two Sorghum genotypes (resistant cultivar RTx2783 and susceptible cultivar BCK60) were used. Leaf samples were collected from treated and control plants at 5, 10, and 15 days post sugarcane aphid infestation. Three biological replicates were performed for all treatment and time combinations. RNA was sequenced using the Illumina Hiseq 2500 platform to generate 100 bp single end reads. The accession numbers and sample information were listed in Table S1. The differential expression of SbCys genes were investigated by Hisat2 (http:/kim-lab.org/), Htseq (http://www.htseq.org/), and DESeq2 (R package) based on the RNA-seq data (Wen, 2017). The p ≤ 0.05 and —logFC—≥ 1.5 were set as the cut-off criterion.

Plant materials and treatments

Seed of Sorghum (Sorghum bicolor L. cv. Jinza 35) were surface sterilized (15 min in 4% NaClO), washed with distilled water several times, and transferred to moist germination paper for 3 days in an incubator at 25 °C. These seedlings were grown in holes of foam floating plastic containers (30 seedlings per container) with constant aeration in Hoagland solution in a growth room with 14 h/30 °C light and 10 h/22 °C dark regime. The nutrient solution was routinely changed every 3 days. At the three-leaf stage (the juvenile phase (Hashimoto, Tezuka & Yokoi, 2019), abiotic stresses including ABA, salinity, and dehydration treatments were initiated according to the procedures described in previous reports (Dugas et al., 2011; Wang et al., 2012; Yan et al., 2017). The plants were transferred quickly to the nutrient solution containing 0.1 mM ABA (dissolved in ethanol), 5 µL ethanol (control for ABA treatment), 250 mM sodium chloride (NaCl), or 15% (W/V) polyethylene glycol (PEG) 6,000. The central part of flag leaves from randomly selected Sorghum plants were harvested respectively at 0, 12, and 24 h post-treatment per trial, and immediately frozen in liquid nitrogen and then stored at −80 °C prior to RNA isolation. For each treatment at a given time, three biological replicates were used. The leaf samples of 10 plants came from the same container for one biological replicate. That is, three containers were used for three biological replicates respectively.

RNA extraction and qRT-PCR analysis

Total RNA of 100 mg leaf samples was isolated using the “TaKaRa MiniBEST Plant RNA Extraction” Kit (TaKaRa, Dalian, China) following the manufacturer’s instructions. Purity and concentration of RNA samples were evaluated by measuring the A260/A230 and A260/A280 ratios. In order to digest the genomic DNA, the RNAs were treated with RNase-free DNase I. Reverse transcription was performed according to the kit instructions (Promega, Madison, USA). Primer pairs for qRT-PCR analysis were designed by Primer3Plus program (http://www.bioinformatics.nl), and were shown in Table S2. A 20 µl reaction volume containing 0.4 µl of each primer (forward and reverse), 2 µl 10-fold diluted cDNA, 7.2 µl of nuclease-free water, and 10 µl of GoTaq® qPCR Master Mix (Perfect Real Time; Promega). PCR reaction included one cycle at 95 °C for 3 min, followed by 39 cycles of 95 °C for 15 s, 60 °C for 30s, and 72 °C for 20s. The reactions were conducted using the CFX96 Real-Time PCR Detection System (Bio-Rad Laboratories, Inc.). Three independent biological replicates and two technical replicates of each sample were performed. Gene-specific amplification of both reference and cystatin genes were standardized by the presence of a single, dominant peak in the qRT-PCR dissociation curve analyses. All data were analyzed by CFX Manager Software (Bio-Rad Laboratories, Inc.). The efficiency range of the qRT-PCR amplifications for all of the genes tested was between 91% and 100%. The average target (SbCys) cT (threshold cycle) values were normalized to reference (β-actin) cT values. The fold change between treated sample and control was calculated using the slightly modified 2−(ΔΔCt) method as described by Kebrom, Brutnell & Finlayson (2010). A probability of p ≤ 0.05 was considered to be significant.

Results

Identification and analysis of SbCys genes

To extensively identify all of SbCys family members in Sorghum, we constructed a Sorghum-specific HMM for the SbCys domain to scan the Sorghum genome, and 22 gene candidates were identified. After removing the repetitive and/or incomplete sequences, the rest of SbCys sequences were submitted to Pfam (http://pfam.xfam.org/) and SMART (http://smart.embl-heidelberg.de/) to confirm the conserved domain. Finally, a total of 18 non-redundant SbCys proteins were identified and were serially renamed from SbCys1 to SbCys17 according to their location and order in chromosomes. Gene names, gene IDs, chromosomal locations, amino acid numbers, protein sequences, and annotations assigned to GO terms of the identified SbCys proteins were listed in Table S3. The average length of these SbCys proteins was 148 amino acid residues and the length mainly centered on the range of 105 to 240 amino acid residues.

Chromosome distribution analysis showed that the number of SbCys genes on each chromosome is different (Fig. 1). Chromosome 1 had the greatest number of SbCys genes (9 genes), followed by chromosomes 9 and 3 (4 and 3 genes, respectively). Chromosomes 2 and 4 had just one SbCys gene, whereas chromosomes 5, 6, 7, 8, and 10 had no SbCys genes.

Figure 1 Chromosome localization of SbCys genes.

Chromosome number is indicated at the top of each bar. The label for the size of chromosome is shown.

Gene structure analysis of SbCys genes

The analysis of exon-intron structure can provide useful information about the gene function, organization, and evolution of multiple gene families (Xu et al., 2012). Schematic structures of SbCys genes from Sorghum were obtained using the GSDS program (Fig. 2). Among the SbCys genes, more than half (12, 66.7%) were intronless, three genes (SbCys11, SbCys15, and SbCys16) had one intron, two genes (SbCys14 and SbCys17) had two introns, and one gene (SbCys10) had three introns. These six SbCys genes with one or more introns were clustered into one clade, suggesting the evolutionary event may affect the gene structure (Altenhoff et al., 2012).

Figure 2 Phylogenetic relationship and gene structure of SbCys genes.

A phylogenetic tree was constructed using MEGA X by the maximum likelihood method with 1000 bootstrap replicates. Exon/intron structures were identified by online tool GSDS. Lengths of exons and introns of each SbCys genes were exhibited proportionally. Exons and introns are shown by blue bars and black horizontal lines, respectively.

Sequence alignment, protein motifs analysis, and structural predication of SbCys

Alignments of SbCys sequences were carried out to search for amino acid variants that could lead to differences in their inhibitory capability for cysteine proteases. The results were shown in Fig. 3A. N-terminal and C-terminal extensions with varying lengths that presented in several SbCys proteins were not displayed in the comparison. These predicted structures shared many identical residues including α-helix and the four β-sheets (β2-5) (Fig. 3A). Analysis of conserved motifs of SbCys proteins also revealed that some typical conserved motifs could be detected in most SbCys proteins, such as motif 1, 2, 3, and 4. These motifs formed a fundamental structural combination (Figs. 3B and 3C). Motif 1 was conserved in the central loop region with a consensus sequence of “QxVxG” and could be detected in most SbCys proteins, which played an important role in the inhibitory capacity of cystatins towards their target cysteine proteases (Meriem et al., 2010). Motif 2 contained a particular consensus sequence ([LVI][GA][RQG][WF]AV) that conformed to a predicted secondary α-helix structure (Martinez et al., 2009). The other two typical motifs for SbCys proteins, motif 3 (V[WY][EVG]KPW) and motif 4 ([RK]xLxxF), which were firstly described in tobacco (Zhao et al., 2014), were also detected in most SbCys proteins, indicating their conserved and common role in both dicots and monocots. Motif 5 existed only in 3 SbCys family members (SbCys5, SbCys8, and SbCys15). Details of the 5 conserved motifs were shown in Fig. S1.

Figure 3 The amino acid alignment and conserved motifs distribution of SbCys.

(A) The locations of the secondary structures (α-helix and β-sheets) were included. The main cystatin conserved motifs are in black boxes. The strong and weak conservative changes in amino acids are marked by dark gray and light gray font, respectively. (B) The motifs were identified by MEME. Each motif was represented by one color box. (C) Conserved protein motif 1 (QxVxG), motif 2 ( LARFAV and G-residue), motif 3 (W-residue), motif 4 ([RK]xLxxF), and motif 5(P-residue) presented in the variable region of cystatin genes.

The predicted three-dimensional structures of the Sorghum cystatins were established using the SWISS-MODEL program based on the known crystal structure of OC-I and SiCYS (Fig. 4). Although these structures were predicted with variable degrees of accuracy, all of Sorghum cystatins shared similar protein structure with rice OC-I (Fig. 4A), excepting SbCys10 that shared similar protein structure with SiCYS (Fig. 4B). In additon, SbCys14 showed a significant variation in its predicted three-dimensional structures, might due to an extra α-helix that existed in the C-terminal extension of SbCys14. Two important motifs (the conserve QxVxG motif and W residue) of Sorghum cystatins involved in the interaction with the target cysteine enzymes were also shown in Fig. 4. The predicted structure of SbCys13 showed some distortions in the region of the β2 sheet, probably due to the insertion of a methionine in the first position of the conserved QxVxG motif.

Figure 4 The three-dimensional structure prediction of Sorghum cystatins.

(A) The three-dimensional structures of SbCys proteins were predicted using the automated SWISS-MODEL program with OC-I as a template. (B) The three-dimensional structure of SbCys10 was predicted using the automated SWISS-MODEL program with SiCYS as a template. Two important motifs involved in the interaction with the target enzymes are indicated: the reactive site (asterisks) and W residue (crosses).

Phylogenetic analysis of SbCys genes

The cystatin gene family is highly conserved in both monocots and dicotyledons (Martinez & Diaz, 2008). To investigate the phylogenic relationships of SbCys proteins to other known plant cystatins, a multiple sequence alignment of SbCys sequences to the sequences from Arabidopsis, rice, and barley was conducted by the ClustalW program. As showed in Fig. 5, these cystatins were categorized into three groups, including Group I, Group II, and Group III. A total of 21 cystatins were classified to Group I and 6 cystatins from Sorghum. Group II contained 7 cystatins, only one cystatin from Sorghum. The remaining 21 proteins were assigned to Group III and 11 SbCys proteins fell into this group. In addition, some bootstrap values in the phylogenetic tree were low, suggesting that high sequence differentiation in these cystatins occurred. Microsynteny analysis indicated that one orthologous gene pair was identified in the cross of barley and Sorghum, rice and Sorghum, respectively, while no orthologous gene pair between Arabidopsis and Sorghum was found (Fig. S2). These data indicated that SbCys genes were more closely related to rice and barley than Arabidopsis. Interestingly, a pair of SbCys genes (SbCys2-1 and SbCys2-2) was involved in the tandem duplication event in Sorghum (Fig. S2). Analysis of duplicated SbCys genes showed that the Ka/Ks ratio far less than 1, varying from 0.0976 to 0.5679 (Table S4), indicating that negative selection occurred in the duplication event.

Figure 5 Phylogenetic relationships of the cystatins from Arabidopsis, rice, barley and Sorghum.

The phylogenetic tree was constructed by MEGA X with the maximum likelihood method. The numbers at the nodes indicate the bootstrap values. Gene names with black, red, and blue represented Group I, Group II, and Group III, respectively.

Promoter analysis of SbCys genes

In order to obtain useful information on the regulatory mechanism of cystatin gene expression, the 1.5 kb upstream sequences from the translation start sites of SbCys genes were submitted into PlantCARE database to detect the cis-elements. Various putative plant regulatory elements in the promoter region of SbCys genes were shown in Fig. 6 and Table S5. Several potential regulatory elements involved in stress-related transcription factor-binding sites were found, including G-box, W-box, TC-rich repeats, MBS, heat shock elements (HSEs), and ABA-response element (ABRE). The identified SbCys genes possessed at least 1 stress-response-related cis-element, suggesting that the expressions of SbCys genes were related to the biotic and abiotic stresses. All of SbCys genes had one or more G-box with the exception of SbCys9, implying that these SbCys genes could be induced by light stress. 14 SbCys genes possessed MBS element, ABRE element was found in 12 SbCys genes, HSE element was located in 10 SbCys genes, and TC-rich repeats and W-boxes were located in 8 genes. In addition Skn-1_motif was conserved in the promoter regions of most SbCys genes, indicating these genes were associated with the regulation of seed storage protein gene expression (Strömvik & Fauteux, 2009). The high diversity of the cis-acting elements suggested that these SbCys genes might have a wide range of functional roles and could be involved in multiple stress responses and growth and development progress (Zhang, Liu & Takano, 2008).

Figure 6 The distribution of cis-elements in the 1.5 kb upstream promoter regions of SbCys genes.

The cis-elements in the promoter region of SbCys genes were predicted using PlantCARE database ( http://bioinformatics.psb.ugent.be/webtools/plantcare/html/). Different cis-elements were represented by different shapes and colors.

Protein interaction network of SbCys proteins

In this study, the interactions of the SbCys proteins were investigated in an Arabidopsis association model using STRING software. As shown in Figs. 7, the interaction network of cystatins showed a complex functional relationship. AtCys2 (corresponding to SbCys12) interacted with stress related proteins (AT1G56280, AT3G19580, AT5G67450, and AtCys1) and growth and development related proteins (AT1G63100 and AT5G04340), AtCys1 (corresponding to SbCys11, 15, 16, and 17) interacted with some vacuolar-processing enzyme which involved in processing of vacuolar seed protein precursors into the mature forms, and AtCys5 (corresponding to SbCys1, 2-1, 3, 4, 5, 6, 7, 8, 9, and 13) interacted with several lipid-transfer proteins (AT1G07747, AT1G52415, AT2G16592, AT3G29152, and AT4G12825). The results suggested that cystatins might be associated with many biological processes by protein interactions, such as pollen development, stress responses, and seed maturation (Wang et al., 2012).

Figure 7 The interaction networks of SbCys proteins according to the orthologs in Arabidopsis.

Functional interacting network models were integrated using the STRING tool, and the confidence parameters were set at a 0.40 threshold. Homologous genes in Sorghum and Arabidopsis are shown in black and red, respectively.

Expression profile of SbCys genes in different Sorghum tissues

To obtain the spatial and temporal expression patterns of all SbCys genes, RNA-seq data (ERP024508) were downloaded to explore the expression levels of SbCys genes in different tissues including root, stem, seedling, pollen, endosperm, embryo, inflorescence (1–5 mm, 1–10 mm, and 1–2 cm), and pericarp. As shown in Fig. 8 and Fig. S3, most SbCys genes were expressed in one tissue at least, except for SbCys13, which were barely expressed in any tissue. The expression patterns of SbCys genes were significantly different between reproductive tissues and vegetative tissues, such as SbCys2-1, SbCys3, SbCys4, SbCys5, SbCys7, SbCys9, SbCys12, and SbCys17, which showed relatively higher expression levels in reproductive tissues including pollen, endosperm, embryo, and pericarp than in vegetative tissues, while the expression of SbCys7 and SbCys15 were higher in vegetative tissues than in reproductive tissues. It was worth noting that the majority of SbCys genes had lower expression levels during inflorescence development excepting SbCys17 which displayed a higher expression pattern.

Figure 8 Hierarchical clustering of the expression profiles of SbCys genes in different tissues.

Different tissues are exhibited below each column. Root, shoot, and whole organism belonged to vegetable tissues were collected at 14 days after Sorghum seed germination. Reproductive tissues included embryo , endosperm and pericarp were collected at 20 days after pollination; pollens at booting stage; inflorescences based on sizes: 1–5 mm, 5–10 mm, and 1–2 cm. Log transform data was used to create the heatmap. The scale bar represented the fold change (color figure online). Blue blocks represented the lower expression level and red blocks represented the higher expression level.

Expression of SbCys genes under biotic stresses

To gain insight into the potential roles of SbCys genes in response to Bipolaris sorghicola infection and sugarcane aphid infestation, the relative expression patterns of these genes were investigated by using the public transcription data from NCBI SRA database (DRP000986 and SRP162227, respectively). As shown in Fig. 9 and 10, the expression patterns of SbCys genes were different under the two biotic stresses. In response to Bipolaris sorghicola infection, seven SbCys genes were induced and only 2 genes (SbCys12 and SbCys13) were suppressed in the infected Sorghum leaves compared with control (Fig. 9A). However, under aphid infestation, four SbCys genes (SbCys4, SbCys10, SbCys11, and SbCys14) were up-regulated and 3 genes (SbCys1, SbCys3, and SbCys17) were down-regulated relative to control in the susceptible Sorghum line (BCK60). In the resistant Sorghum line (RTx2783), only two SbCys genes (SbCys4 and SbCys11) were induced, and the rest were barely expressed in Sorghum leaves with aphid infestation (Figs. 9B and 10). These results might suggest that SbCys genes played different roles in responding to pathogen infection and aphid infestation.

Figure 9 Hierarchical clustering of the expression profiles of SbCys genes under biotic stresses.

(A) The expression changes in SbCys genes at 0, 12, and 24 h with Bipolaris sorghicola infection. (B) The expression changes of SbCys genes at 5, 10, 15 days with sugarcane aphid infestation. Log transform data was used to create the heatmap. The scale bar represents the fold change (color figure online). Blue blocks indicate low expression and red blocks indicate high expression (color figure online).

Figure 10 Expression profiles of SbCys genes at 5, 10, and 15 days with sugarcane aphid infection.

Expression profiling of SbCys genes under abiotic stresses

We also investigated the expression of SbCys genes in response to various abiotic stresses including dehydration, salt shock, and ABA (Fig. 11). Under dehydration stress, seven SbCys genes (SbCys4, SbCys5, SbCys6, SbCys9, SbCys10, SbCys11, and SbCys17) were induced to present a significant up-regulation from 0 to 24 h, while the expressions of SbCys2-1, SbCys12, SbCys15, and SbCys16 were decreased. Furthermore, the expressions of 4 SbCys genes (SbCys1, SbCys3, SbCys8, and SbCys14) displayed an up-down trend from 0 h to 24 h (Fig. 11A). With salt shock treatment, the expressions of SbCys2-1, SbCys3, SbCys4, SbCys8, SbCys10, and SbCys11 were significantly up-regulated at all treatment time points, whereas SbCys16 showed a significant down-regulated trend (Fig. 11B). In addition, SbCys6, SbCys13 SbCys14, SbCys15, and SbCys17 showed up-down expression trends, but SbCys5 displayed a down-up expression pattern (Fig. 11B). After exogenous ABA treatment, the expressions of 4 SbCys genes (SbCys2-2, SbCys3, SbCys4, and SbCys7) were significantly up-regulated at three time points, but 9 genes (SbCys1, SbCys2-1, SbCys5, SbCys8, SbCys10, SbCys11, SbCys13, SbCys14, and SbCys17) were down-regulated. Additionally, SbCys12, SbCys15, and SbCys16 displayed up-down expression trends (Fig. 11C). Interestingly, all SbCys genes were up-regulated in response to one or two stresses except SbCys4 that was significantly induced under dehydration, salt, and ABA stresses, suggesting that SbCys4 might play an important role in response to different stress responses.

Figure 11 Expression patterns of SbCys. genes under (A) dehydration (PEG 6,000) treatment, (B) salt shock (NaCl) treatment, and (C) ABA treatment.

qRT-PCR was used to investigate the expression levels of each SbCys gene. To visualize the relative expression levels data, 0 h at each treatment was normalized as “1”. * indicated significant differences in comparison with the control at p ≤ 0.05. ** indicated significant differences in comparison with the control at p ≤ 0.01.

Discussion

Plant cystatins are a group of intrinsic small proteins, whose members play important roles in diverse biological processes and stress responses (Martinez et al., 2016; Meriem et al., 2010). Recently, a large number of sequence data from different plant species have been uploaded in GenBank, which provide convenience for us to describe their characteristics, and several cystatins families have been identified from plants, such as rice, soybean, and wheat (Wang et al., 2015; Dutt et al. 2012; Yuan et al., 2016). However, little is known about the cystatin family in Sorghum. In the present study, we identified 18 SbCys genes from the Sorghum genome. The number was less than that of B. distachyon genome, where 25 BdCys members were identified (Subburaj et al., 2017). The 18 members in Sorghum was a larger number than found in rice (11 genes) and Arabidopsis (7 genes) (Wang et al., 2015), but was similar to soybean (20 members) (Yuan et al., 2016). The difference on the cystatin number might reflect the adaptation of plants to environment.

The identified SbCys genes were unevenly distributed on chromosomes 1, 2, 3, 4, and 9, and half of them were distributed on chromosome 1 (Fig. 1). The uneven distribution of cystatin genes in chromosomes was also found in the B. distachyon genome and the Oryza sativa genome (Subburaj et al., 2017; Wang et al., 2015). This phenomenon might be due to the tandem duplication events of cystatin genes on the chromosomes (Li et al., 2017a; Li et al., 2017b). Several tandem duplication events occurred at chromosomes 1 of the B. distachyon genome (Subburaj et al., 2017). Two tandem duplication events (OsCys4/OsCys5 and OsCys6/OsCys7) were found among OsCys genes, and existed in chromosomes 1 and 3 (Wang et al., 2015). One tandem duplication event (SbCys2-1/SbCys2-2) occurred at chromosome 1 of the Sorghum genome (Fig. S2). Eighteen SbCys genes were divided into three groups based on phylogenetic analysis (Fig. 5). Some conserved motifs among SbCys proteins had been identified by the alignment of the amino acid sequences (Fig. 3). However, the conservation was accompanied with the differences in some important amino acids indicated that SbCys family members might undergo a complex evolutionary history. The variation of crucial amino acids of cystatins might have a significant influence on their respective functions (Tremblay, Goulet & Michaud, 2019). For example, the QxVxG motif could directly enter and interact with the active site of targeted enzymes. The motif was conserved in all SbCys proteins with the exceptions of 5 cystatins (SbCys1, SbCys6, SbCys8, SbCys9, and SbCys13) that were partially modified by the insertion or variation in important residues (Fig. 3A). Furthermore, three SbCys proteins (SbCys8, SbCys9, and SbCys13) showed significant variations with other Sorghum cystatins in their predicted three-dimensional structures (Fig. 4). The variations in vital amino acid residues might result in the change in cystatin inhibitory action (Tremblay, Goulet & Michaud, 2019). In addition, two novel motifs, motif 3 (V[WY][EVG]KPW) and motif 4 ([RK]xLxxF), firstly described in tobacco (Zhao et al., 2014), were also identified in the C-terminalin of many SbCys proteins. The contribution of the two new motifs to cystatin inhibitory action needs to be further studied.

During past decades, plant cystatins were reported to play essential roles in inhibiting endogenous and exogenous cysteine proteases activities during seed development (Tremblay, Goulet & Michaud, 2019). In the present study, as revealed by RNA-seq data analysis (Fig. 8 and Fig. S3), the expression levels of several SbCys family genes were higher in reproductive tissues than in vegetative tissues, which were consistent with the reports that most cystatins were specifically expressed in developing seeds and played a role in seed development (Dutt et al., 2010; Zhao et al., 2014). Moreover, promoter analysis showed that the highly expressed SbCys genes in reproductive tissues possessed endosperm expression-related cis-elements (Skn-1 and GCN4_motif) (Fig. 6 and Table S5). Our protein interaction prediction results also showed that several SbCys proteins could interact with many functional proteins (e.g., growth and development related proteins, vacuolar-processing enzyme, and lipid-transfer proteins) (Fig. 7), implying these cystatins were involved in regulating the gene expression of cereal grain storage proteins (Diaz-Mendoza et al., 2016).

Plant cystatins are involved in various biotic stress responses and probably act as defense proteins against pest infestation and pathogen infection (Meriem et al., 2010). At present, some cystatins with insecticidal activity have been isolated from many plants, such as barley, tomato, and potato (Rasoolizadeh et al., 2017; Siddiqui, Khaki & Bano, 2017; Velasco-Arroyo et al., 2018; Goulet, Sainsbury & Michaud, 2020). Several cystatins having antifungal activities were also isolated from taro, cacao, and wheat (Christova et al., 2018; Pirovani et al., 2010; Chen et al., 2014). Although studies on insecticidal and antifungal activity of plant cystatins have been well established in vitro, the knowledge about their roles in plants in response to biotic stresses is limited. To explore the properties of SbCys genes responding to pest infestation and pathogen infection, we conducted the analysis on the expression patterns of SbCys genes. The results showed that the expressions of most SbCys genes were induced during Bipolaris sorghicola infection, suggesting these cystatins played functions in inhibiting exogenous cysteine proteases secreted by pathogens to infect plant tissues (Fig. 9A). Interestingly, for sugarcane arthropods infestation, only two genes (SbCys4 and SbCys11) were up-regulated significantly in susceptible and resistant Sorghum lines (Figs. 9B and 10). These differential expression patterns between SbCys genes might suggest that some of them had evolved to inhibit specific cysteine proteinases. The exact roles of these SbCys genes in insecticidal and antifungal activity in vivo are worthy to be explored in further study.

Cystatin genes are also involved in various abiotic stress responses in plants. In Arabidopsis, the expression levels of AtCYS1 and AtCYS2 were enhanced by high temperature and wounding stresses (Hwang et al., 2010). AtCYSa and AtCYSb were also induced by different abiotic stresses, e.g., salt, drought, oxidation, and cold stresses (Zhang, Liu & Takano, 2008). Velasco-Arroyo et al. (2018) reported that the silence of barley HvCPI-2 and HvCPI-4 specifically modified leaf responses to drought stress. Wang et al. (2015) observed the significant change in the expression levels of several rice OsCYS genes under cold, drought, salt, and hormone treatments. In the present study, most SbCys genes were found to have positive or negative responses to dehydration, salt, and ABA stresses. Moreover, the interaction results showed that most cystatins could interact with stresses-related proteins, implying that the cystatins played critical roles in response to diverse stress conditions. Notably, the expression of SbCys4 was significantly up-regulated under three stress conditions (Fig. 11), suggesting a specific role of SbCys4 in responding to various stress conditions. Promoter analysis indicated that stress-related cis-elements were widespread in the promoter region of these cystatin genes (Table S5), and SbCys4 possessed plenty of stress-related cis-elements, including G-box, ABRE, HSE, MBS, and TC-rich repeats. These results provide an effective reference for the functional verification of the SbCys family genes under abiotic stresses.

Conclusions

In the current study, we identified 18 SbCys family genes in the Sorghum genome through a genome-wide survey. The chromosomal localization, conserved protein domain, gene structure, phylogenetic relationship, as well as the interaction network of these SbCys genes was systematically analyzed, revealing special characteristics of SbCys family genes in Sorghum. The identified SbCys genes displayed an uneven distribution in Sorghum chromosomes. All SbCys genes shared similar exon/intron organization and conserved motifs. Phylogenetic analysis suggested that Sorghum cystatins had higher homology with monocotyledon than dicotyledon. Furthermore, the variation of amino acids in Sorghum cystatin critical active sites suggested that they might undergo a complex evolutionary process and possess structural and functional divergence. The expression profiles of SbCys genes in different tissues indicated that most SbCys genes were involved in plant growth and development. Changes in the expression of SbCys genes under biotic and abiotic stresses indicated that many SbCys genes played important roles in response to unfavorable growth conditions. It is worth noting that the expression of SbCys4 was significantly enhanced under biotic and abiotic stresses, suggesting its unique role in mediating the response of Sorghum to adverse environmental conditions.

Supplemental Information

Supplemental Information 1 Amino acid sequence of conserved motifs in SbCys proteins

Click here for additional data file.

Supplemental Information 2 Microsynteny analyses of cystatin genes among rice (Os), barley (Hv), Arabidopsis (At), and Sorghum (Sb)

Green lines connecting two chromosomal regions indicated syntenic regions between rice and sorghum, barley and sorghum. Red lines denoted tandem duplication in Sorghum chromosome.

Click here for additional data file.

Supplemental Information 3 Expression profiles of SbCys. genes in different tissues

The data represented fold change (logFC value).

Click here for additional data file.

Supplemental Information 4 The accession number and sample information of RNA-seq data

Click here for additional data file.

Supplemental Information 5 Primer sequences used for qRT-PCR.

Click here for additional data file.

Supplemental Information 6 Gene name, chromosomal location, amino acid sequence, and functional annotations of identified Sorghum cystatins

Click here for additional data file.

Supplemental Information 7 The Ka./Ks analysis of SbCys genes

Click here for additional data file.

Supplemental Information 8 The number of promoter elements of each SbCys gene

Click here for additional data file.

Additional Information and Declarations

Competing Interests

Author Contributions

Data Availability

The authors declare there are no competing interests.

Jie Li conceived and designed the experiments, prepared figures and/or tables, authored or reviewed drafts of the paper, and approved the final draft.

Xinhao Liu performed the experiments, analyzed the data, prepared figures and/or tables, and approved the final draft.

Qingmei Wang and Junyan Sun analyzed the data, authored or reviewed drafts of the paper, and approved the final draft.

Dexian He conceived and designed the experiments, authored or reviewed drafts of the paper, and approved the final draft.

The following information was supplied regarding data availability:

The raw data are available in the Supplemental Files.

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
