# Peer review of "Genome-wide identification and analysis of cystatin family genes in Sorghum (Sorghum bicolor (L.) Moench)"

_PeerJ, doi:10.7717/peerj.10617_

## Round 0.1 · original submission · Major Revisions

1.Most of the expression profiles in the article rely on transcriptome data, but too little information about transcriptome is introduced. Such as data reliability, sampling methods, etc.The content of transcriptome sequencing needs to be described in detail in the method. How many replicates do you use for your transcriptome? The transcriptomic analysis is based on your replicates of samples among different treatments. You should highlight how many replicates do you test and show the repeatability error in Results section to make sure this experiment is reliable.
2.Why are these gene family members not renamed, but rather gene IDs?
3.What is the basis for the selection of the internal reference gene of QRT-PCR?
4. The author's material method section has lost a lot of important information. The sampling period to test should have some explaination about why choosing these days to get and describe the characteristics about all samples during this period. It is recommended to supplement physiological data.
5.The number of genes analyzed by QRT-PCR should increase, especially those using transcriptome analysis. Histograms should be used and analyzed for significance.
6.The final conclusion and purpose of the article are too vague. Through bioinformatics and expression profiling, which genes are ultimately considered important? No description.

·

Basic reporting

No

Experimental design

The present study "Molecular characterization and expression profiling of cystatin family genes reveal their high evolutionary conservation and functional divergence in sorghum (Sorghum bicolor L.)" attempt to underpin the evolutionary conservation and functional divergence of cystatin family genes in Sorghum bicolor. It is an interesting study and relevant to current focus. This study does highlight cystatin gene characterization evolutionary relatedness and their expression profiles. However, considering the interest of the global readers of this journal, authors must improve the manuscript further. Real time expression analysis can be designed in a comprehensive manner. some more studies like Protein modelling and inhibition sites finding can make it a novel work in Sorghum.

Validity of the findings

Conclusion can be written focusing mainly on the present work carried out instead of generalized Findings. Mention the group of genes fall into different localizations, exon-intron characterization and real time expression analysis, duplication events, phylogeny etc.,

Additional comments

This study does highlight cystatin gene characterization evolutionary relatedness and their expression profiles. However, considering the interest of the global readers of this journal, authors must improve the manuscript further. Real time expression analysis can be designed in a effective manner. some more studies like Protein modelling and inhibition sites finding, finding interacting partners can make it a novel work in Sorghum.

·

Basic reporting

This manuscript studied the cystatin – inhibitor of cysteine protease in Sorghum. Real-time quantitative polymerase chain reaction (RT-qPCR) and some bioinformatic prediction w tools were used to support the functional diversity of SbCys genes related to specific inhibition activity in different tissues.
The introduction doe well organize paragraphs in order to emphasize the necessity of this study. Literature is well referenced and relevant on studied topic.

Experimental design

Methods was described with sufficient detail and experimental/bioinformatics tools used in this research was well defined, relevant and meaningful.

Validity of the findings

The impact and novelty of this manuscript are assessed. Overall, this manuscript was qualified for publication.
Minor comments:
Figure 1: Description in more detail may needed for protein ID
Figure 6,8: Unit for heat map scale.
Figure 7: similar scale for Figure 7A and 7B
Page 10, line 115 – 116: remove space between number and %unit (>75%)

Reviewer 3 ·

Basic reporting

In the manuscript entitled “Molecular characterization and expression profiling of cystatin family genes reveal their high evolutionary conservation and functional divergence in sorghum (Sorghum bicolor L.)”, the authors predicted 18 sorghum cystatin genes (SbCys) with an HMM and analyzed their protein motifs, chromosome localization, cis-elements and expressions. These analyses for 18 SbCys genes are important for sorghum research. In the current manuscript, however, the connection of these analyses is not clear. Adding a schematic representation of putative regulatory flow for the cystatin genes would help readers to their deeper understanding.

Experimental design

Experimental design for identifying 18 SbCys genes with an HMM is clearly written but the authors need to mention the reasons why they chose this method instead of simply applying blast searches nor domain searches.

Validity of the findings

1) The authors describe that SbCys genes have a chromosomal preference in Results (Ll. 167-) and Discussion (Ll. 300-). In a related comparison with O. sativa (Wang W. et al. 2015 Plant Cell Rep), the chromosomal preference looked different between rice and sorghum. How do you interpret these differences? Also, the number of cystatin family genes is larger in sorghum than others (e.g. 7 genes in A. thaliana, 11 genes in O. sativa and 10 genes in N. tabacum). What is the biological and/or evolutional meaning? To check the evolutional duplication, Wang, et al. checked the Ka/Ks ratios between duplicated cystatins. I recommend the authors to refer this work.
2) As for the promoter analysis, the authors mentioned that Skn-1 and GCN4 motifs are abundant. The motif length of Skn-1 is short. Isn’t it possible that the observed abundance is just because of its shortness? Please perform the statistical test. Also, are there any previous reports that Skn-1 or GCN4 act on cystatin genes in any other plants?
3) Fig. 6: How is the expression profile of transcription factors, Skn-1/GCN4 ?

Additional comments

Minor comments:
L. 93 and L. 133: Please describe the parameters for HMMER and the RNA-seq applications.
Supplemental tables: Please add JGI gene numbers. Many sorghum researchers use JGI gene numbers (https://phytozome.jgi.doe.gov/pz/portal.html#!info?alias=Org_Sbicolor).
L. 117-: The explanation of microsynteny analysis is not clear enough.

Fig. 2(a): Please increase the font size
Fig. 2(b): Please remove unnecessary all “-“ columns and make the figure larger.
Fig. 6,7: Please add the explanation of each scale bar.
Fig. S1: Difficult to read the gene names.

L. 56: a-helix ->α
(L. 95: E-value ->e-value)
L. 134: P -> p
L. 138: 10-cm -> 10 cm

---

## Round 0.2 · Major Revisions

I think that although the quality of the manuscript has been improved, the depth is still not enough. The specific reviewers' comments are attached.


·

Basic reporting

The author has addressed all comments and improved the manuscript in well defined manner in grammatical mistakes, references, etc.,

Experimental design

The author has addressed and included modelling of proteins, and interaction maps also. why author feels that results were unsatisfactory for Protein interaction map. Have they tried and crosschecked with other tools like Genemania and cytoscape.

Validity of the findings

All the data of experimental findings, figures, discussion and conclusions were clearly written according to reviewer comments.

Additional comments

If the author feels unsatisfactory with protein interaction studies, they can remove it (it is not mandate, but in future, can try to predict and prove experimentally to validate their findings).

Reviewer 3 ·

Basic reporting

no comment

Experimental design

no comment

Validity of the findings

no comment

Additional comments

The manuscript and figures were widely improved and the authors answered all my concerns. However, there are still some small mistakes. I recommend to fix the following mistakes.



Response: To identify the SbCys genes, two methods (HMM and blast searches) were used in the study. The information was shown in lines 132-147.
It is better to describe your method in “Identification of SbCys family members in Sorghum genome”, not in the other section.

Fig 2 legend: Better to remove (a) and (b) from the legend.

Fig 6: In the Wang B et al. 2018 Genome Res, they released Iso-seq data, too. Can you check the TSSs for 18 SbCys genes and add able to put Fig. 6?

Fig 10: Please carefully put “*” in the figure. Some “*” are slightly shifted. Especially Fig C at KXG22254 gene.

Reviewer 4 ·

Basic reporting

The manuscript shoud be gone through by a native English speaker.

Experimental design

these preliminary data could not support this big title, and authors need to do more experimental works rather than present some rough results if they really want to state the importance of SbCys in Sorghum.

The experimental design and writing logic wait to be improved.

Validity of the findings

no comment.

Additional comments

Dear Editor,

I was very glad to be a reviewer of PeerJ. However, after reviewing this manuscript carefully that entitled ‘Molecular characterization and expression profiling of cystatin family genes reveal their high evolutionary conservation and functional divergence in Sorghum (Sorghum bicolor L.)’, I got little insightful messages from this manuscript.

In my opinion, these preliminary data could not support this big title, and authors need to do more experimental works rather than present some rough results if they really want to state the importance of SbCys in Sorghum. Nowadays, transient expression assays are easy to do and transgenic Arabidopsis is easy to be generated in any scientific institutions. In addition, the experimental design and writing logic wait to be improved.

As some details, I fully agree with former reviewers. For example, using gene ID as gene name makes it difficult to link the context. No one use ID number as their name.

Considering some efforts had been made by former reviewers, I suggest authors to supply some practical evidences proving the potential function of these SbCys genes in plant responses to drought or other abiotic stresses.

·

Basic reporting

The manuscript entitled “Molecular characterization and expression profiling of cystatin family genes reveal their high evolutionary conservation and functional divergence in sorghum (Sorghum bicolor L.)”, summarized the systematic information about 18 sorghum cystatin genes (SbCys) and their expression and bioinformatics tools like protein motifs, chromosome localization, cis-elements.The authors focused on the cystatin genes and their possible regulatory mechanism that can provide the important information as well for the new researchers. please consider the following concerns and required to be addressed for brief description of the manuscript.

1. In introduction section, I would like to suggest that the authors need to enrich this section by describing the current status or response against biotic stress of cystein genes and phytohormones if latest literature is available.
2. The regulation of gene expression and their responses to various stresses or with other species may also provide important clues and strengthen the author hypothesis please highlight in few sentences by providing fresh references in this section. If possible.

3. The author should clearly conclude the key results of the manuscript, how these genes are important and they can be beneficial/significance and what regard.

4. Please correct the grammar mistakes in the entire manuscript with a focus to the details of spelling, mistypings, italic and punctuation. There are small errors and enormous numbers of inaccuracy and negligence, please focus on too long or complicated sentences, which often describe the poor meaning.

Experimental design

Experimental design used in this manuscript is overall good. though, i have few comments for the author’s consideration hoping it will further improve this manuscript. Please fix the following comments.

3. In M&M section, my particular concern is about the absence of significances and probabilities (P>…) for differences in the presented data. How many biological replicates (plants) were used for each of gene expression analysis?
4. In line 189, Three independent biological repeats, The data collected three independent experiments” is unclear. If three plants were grown together and used as three biological replicates, this means one experiment. ‘Three independent experiments’ mean that plants were grown independently (simultaneously or consequently one-by-one) but number of used plants (biological replicates) must be indicated. Please clarify. Please also include ‘number of technical replicates’ for qPCRfor example ‘qPCR runs were repeated twice (or in triplicates)’. The phrase on L165 probably means the number of technical replicates but has to be clarified: “…from three independent PCR amplifications”. This will indicate for the accuracy of the used BioRad qPCR equipment and a Cycler Altogether, the received results with used number of biological and technical replicates have to be statistically treated with appropriate level of probability and significance.
5. The M&M about abiotic stresses, ABA treatment is written absolutely incompletely, as a draft, where most of details are missed. This makes impossible to reproduce experiments by anybody, There are a lot of inaccuracies and negligence in this sub-section (as well as in entire manuscript) and MUST be fixed by Authors prior the submission. . Please insert sufficient information about hydroponics details, like how many plants/seedlings per container were used? How many containers in one experiment simultaneously or there were several experiments in time (see point 1)? Which water was used for hydroponics: tap-water, RO-water, distilled water, milli-Q water? Please provide either reference or full composition. The Growth solution was changed (how often?) or not? Aerated/non-aerated? for example: on the mesh, floating in holes of foam on the surface, in holes of the lid or another? All these and other sufficient details must be added in this sub-section.
6. Following the comments please provide the description of how all stresses and hormones were applied to the plants/seedlings? Please make corrections accordingly. Firstly, ‘20% PEG 6000’ is not ‘drought’ but dehydration caused by absorption of water with PAG. Please replace it for ‘dehydration’ or ‘simulation of drought’. How PEG 6000 was applied to the hydroponics? What concentration, volume, quick/slow, single/several steps of adding to the hydroponics?
7. Salinity is very different stress. Please indicate how young plants were transferred from regular Growth solution (without NaCl) to those with 250 mM NaCl? If so strong salt stress was applied to the young plants suddenly, in one step, it definitely will cause ‘osmotic shock’ and ‘cell plasmolysis’(see and use a recent Review: J Exp Bot 2013, 64, 119-127). How Authors manage their response with salt stress and salt shock?
8. ABA is water insoluble but can be dissolved in ethanol and some other solvents. Please inserte your information in the text how ABA is dissolved and how applied to the plants? It was an adding of stock solutions with each of the hormones to the original Growth solution or a transferring of plants into a new Growth solution containing tested hormone?
9. How much leaf tissue was used for RNA extraction, What volume and how pure RNA was received? How the quality of RNA was checked (for example, running on the gel) or not? Please indicate, if DNase treatment was included in the RNAiso TM Plus kit or not? Please provide more details about cDNA Synthesis kit (Promega, Madison, USA). for example, how much purified RNA was used for cDNA synthesis and how much cDNA volume was received in the end. Did Authors used non-diluted or diluted (indicate a number of folds) cDNA for qPCR analyses.
10. Please clarify, whole flag leaf or which part of leaf were collected during sampling for RNA extraction, not well clear.
11. My minor concern is that genes name should be mentioned rather using ID, the gene name may be given according to chromosome nomenclature.
12. Did the author checked the other reference genes and their expression variation for qRT-PCR?

Validity of the findings

The findings of this manuscript is interesting and can be considered for publication. please improve the final manuscript with the following suggestions prior to submission.

---

## Round 0.3 · Major Revisions

Reviewers have provided some basic feedback, yet based on the brevity of the comments it may be necessary to seek additional reviews. However, based on an initial read of the manuscript some areas of grammar may need attention. I have also included a markup PDF and have highlighted some areas of concern, it is not my role to correct these issues. PeerJ does not offer copyediting, please ensure that the English language in this submission meets journal standards; this includes
use of clear and unambiguous text which is grammatically correct, and conforms to professional standards of courtesy and expression.

The manuscript does require some revision as a basic bioinformatic survey derived from previous datasets. The reasoning for perceived locations of genes in the genome should be compared based on evolutionary evidence since other reference genomes are used and referred to. Because the identification and function of the genes with respect to tissue, stresses, and perhaps developmental stages a more recognized annotation is needed.

A common annotation used are gene ontology (GO:) terms. Journal manuscripts are often scanned by text-mining software that locates and extracts core data elements, like gene function. Adding standard ontology terms, such as the Gene Ontology (GO, geneontology.org) or others from the OBO foundry (obofoundry.org) can enhance the recognition of your contribution and description. This will also make human curation of literature easier and more accurate. None of this was visible.

Also missing were clear connections to the data for the reader to refer to. It seems that some of this might be deciphered from Table S3. However, in building annotations of the sequences to ontologies the appropriate GO: terms should be included withing the appropriate tables and mentioned within the manuscript. I will place the manuscript at this stage a requiring moderate revision, and upon its return may seek additional reviews based on the clarity of the revised version. Apologies for the delays in getting a review back and we hope to see an improved revision. Thank you for your understanding.

·

Basic reporting

1. Mention vegetative tissue not vegetable all through the MS.

2. Mention the classification and distribution of Cystatins in Introduction.


3. Cite all the latest references pertaining to this work. The author has not mentioned the Genome wide analysis of Cytstatin in Brachypodium (2017). (https://www.ncbi.nlm.nih.gov/pmc/articles/PMC5423411/). Mentioned in Discussion.

Experimental design

No comment

Validity of the findings

No comment

Reviewer 3 ·

Basic reporting

In this manuscript, the authors have applied various functional annotations for 18 predicted sorghum cystatin genes (SbCys) using publicly available tools and data, rather than revealing cutting-edge facts with a meaningful perspective. Their approach is quite common and classical. Because of this reason, this manuscript seems to be interesting for only very limited researchers in sorghum field. I strongly recommend to modify the story of the manuscript.

Experimental design

Although the methodologies do not contain any clear mistakes, the approaches were quite common and also old. Therefore, not so highly valuable information is presented in each figure, except Fig. 11.

Validity of the findings

In the manuscript, the authors only focus on 18 SbCys genes. Thus, the title “Genome-wide identification” may be a little inadequate. They do not show original genome-wide data, either. If this manuscript is focusing on the genome-wide functional annotation, the authors should show some original genome-wide data, such as ChIP-seq and RNA-seq data. If the authors focus on public data, then authors should take care of wider data in sorghum.

Additional comments

Please confirm the usage of following biological terms:
- “Sorghum bicolor L.” should be followed by “Moench”?
- Vegetable tissue -> vegetative tissue?
- “infected” and “infested” were sometimes used incorrectly.

Reviewer 6 ·

Basic reporting

The manuscript entitled “Genome-wide identification and analysis of cystatin family genes in Sorghum (Sorghum bicolor L.)” by Li, et al.; the authors performed a systematic study of the Sorghum cystatins (SbCys) gene family, and they also analyzed the tissue expression patterns as well as their response to abiotic stresses. The data will provide a foundation for further research of the better understanding the roles and regulatory mechanism of Sorghum cystatins in seed development and responses to different stress conditions.

Experimental design

I think the experimental design of this revised version meet the standards of Peer J.

Validity of the findings

no further comment

Additional comments

In this revised version, the manuscript is well-organized and clear to me. I do not have further suggestions on improving the quality of this manuscript.

Only some small gramma or typo issues


Page 1, Abstract, Line32, “one tested tissues” is not correct, please modify it

page 8:RNA samples were collected at 0, 12 and 24 hours, there should be a “,” between 12 and “and”; there’re still many such typos, please double check the whole manuscript, for example line 185 in the same page


Page 11, Line 262, “effect” should be “affect”?

Page 20 line 477, please re-phrase this sentence.


Should the transcriptome data be transformed to the server, so that you can get an accession number before it could be published?

---

## Round 0.4 · accepted · Accept

It appears that attention has been applied to the reviewers' suggestions. The manuscript is currently in an acceptable form, reads fairly cleanly, and is in a form to move forward. I feel the information provided for the SbCys4 and SbCys11 members of the gene family highlight useful information. Congratulations on your efforts.